# Reliability of maximal respiratory nasal pressure tests in healthy young adults

**Jackson C. C de Lima**[1,2]*, **Vanessa R. Resqueti**[1,2], **Ana Aline Marcelino**[1,2], **Jéssica Danielle M. da Fonsêca**[1,2], **Ana Lista Paz**[3], **Fernando A. Lavezzo Dias**[4], **Matias Otto-Yañez**[5], **Guilherme A. F. Fregonezi**[1,2]

1 PneumoCardioVascular Lab/HUOL, Hospital Universitário Onofre Lopes, Empresa Brasileira de Serviços Hospitalares (EBSERH) & Departamento de Fisioterapia Universidade Federal do Rio Grande do Norte, Natal, Rio Grande do Norte, Brasil, 2 Laboratório de Inovação Tecnológica em Reabilitação, Departamento de Fisioterapia, Universidade Federal do Rio Grande do Norte, Natal, Rio Grande do Norte, Brasil, 3 Facultad de Fisioterapia, Universidade da Coruña, A Coruña, España, 4 Departamento de Fisiologia, Universidade Federal do Paraná, Paraná, Brasil, 5 Kinesiología, Universidad Autónoma de Chile, Santiago, Chile

* guilherme.fregonezi@ufrn.br

## Abstract

### Introduction

Sniff nasal inspiratory (SNIP) and expiratory pressure (SNEP) may complement the assessment of respiratory muscle strength. Thus, specifying their reliability is relevant to improving the clinical consistency of both tests.

### Objective

To assess the reliability of SNIP and SNEP in healthy young adults.

### Methods

This cross-sectional study included self-reported healthy aged 18 to 29 years. SNIP was performed using a plug to occlude one nostril, while SNEP was conducted using a face-mask. Participants performed 20 SNIP and SNEP maneuvers with 30-second intervals in between. The intraclass correlation coefficient (ICC), standard error of measurement (SEM), and minimum detectable change (MDC) assessed the reliability of SNIP and SNEP. Analyses were conducted between the highest peak pressure and the first reproducible maneuver in men and women.

### Results

The total sample comprised 32 participants: 16 men and 16 women. The ICC, SEM, and MDC for SNIP maneuvers were 0.994 (95%CI 0.988 to 0.997), 1.820 cmH$_2$O, and 5.043 cmH$_2$O, respectively. For SNEP, these parameters were 0.950 (95%CI 0.897 to 0.976), 6.03 cmH$_2$O, and 16.716 cmH$_2$O. The SNIP and SNEP in men showed ICC of 0.992 (95% CI 0.977 to 0.997) and 0.877 (95%CI 0.648 to 0.957), SEM of 2.07 and 7.66 cmH$_2$O, and MDC of 5.74 and 21.23 cmH$_2$O. In women, SNIP and SNEP presented ICC of 0.992 (95% CI 0.977 to 0.997) and 0.957 (95%CI 0.878 to 0.985), SEM of 1.15 and 6.11 cmH$_2$O, and

**Data Availability Statement:** An excel data will send.

**Funding:** This study was financed in part by the Coordenação de Aperfeiçoamento de Pessoal de

Nível Superior – Brasil (CAPES) – Finance Code 001. GF 316937/2021-5 Conselho Nacional de Desenvolvimento Científico e Tecnológico - CNPq VR - 305960/2021-0 - Conselho Nacional de Desenvolvimento Científico e Tecnológico - CNPq The funders had no role in study design, data collection and analysis, decision to publish, or preparation of the manuscript.

**Competing interests:** The authors have declared that no competing interests exist.

MDC of 3.19 and 16.95 cmH$_2$O. Also, 60% of the highest SNIPs occurred among the 11th and 20th maneuvers in men and women. In men, 55% of the highest SNEPs occurred among the 11th and 20th maneuvers; this value was 50% in women.

## Conclusion

SNIP and SNEP showed excellent reliability. The reliability of SNIP and SNEP in men was good and excellent, respectively, whereas both tests had excellent reliability in women. Also, women reached the highest peak pressure faster than men in both tests.

## Introduction

Non-invasive methods to assess respiratory muscle strength (RMS) emerged in the early 20th century and improved over the decades [1, 2]. Currently, maximum inspiratory (MIP) and expiratory pressure (MEP) assess RMS using mechanical or digital manovacuometers equipped with pressure transducers and connected to an interface (mouthpiece or face mask) [3, 4].

MIP and MEP are reliable and widely used due to their simplicity, practicality, and tolerability. However, performance in these tests may be challenging since they require coordination, collaboration, motivation, integrity of facial muscles, and maintenance of pressure for at least 1.5 seconds [3, 5]. In addition, the variability of MIP and MEP depends if the device used is based on pressure transducers or mechanical gauges [3].

Complementary methods to assess RMS have been developed over the last 20 years [6–8]. For example, sniff nasal inspiratory pressure (SNIP) measures inspiratory muscle strength using short and sharp maneuvers performed with the nose. Although SNIP is slightly different from MIP, it can be used as a complementary RMS assessment, mainly in patients with the neuromuscular disease [9]. Despite scientific advances, few studies investigated the reliability and number of SNIP maneuvers needed to obtain consistent values. Lofaso et al. [7] determined that ten to twenty maneuvers were enough to achieve reliable results, considering that the learning effect continued after the 10th maneuver.

Regarding the assessment of expiratory muscle strength, Morgan et al. [10] described for the first time a method similar to SNIP, the sniff nasal expiratory force. Subsequently, Ichikawa et al. [11] presented the assessment of sniff nasal expiratory pressure (SNEP). Since the technical development, the MIP and MEP tests differ from that of SNIP and SNEP from the air inlet route, the lung volume where the maneuver is carried out, the duration of the test, and possible muscle activation, therefore reliability cannot be extrapolated between tests. Furthermore, gender could influence SNIP and SNEP results as well as it influences other respiratory variables. Preliminary data, from SNIP and MIP comparison, suggested that both tests should be used for early detection of respiratory muscle weakness to avoid over/under diagnoses [12, 13]. The focus on precision rehabilitation requires that tests for evaluating respiratory muscle function be reliable and reproducible with each other. Tests that have low reliability and reproducibility cannot be presented as an evaluation alternative. This is why it is necessary to establish these values for these evaluations. Nevertheless, the reliability and reproducibility of SNIP and SNEP in adults are still unknown. Therefore, this study aimed to analyze the reproducibility and reliability of SNIP and SNEP in healthy young adults.

## Methods

### Participants

A cross-sectional study was conducted using the following inclusion criteria: (1) healthy individuals of both sexes; (2) age between 18 and 29 years; (3) body mass index between 18.5 and 24.9 Kg/m$^2$; (4) no history of respiratory, neuromuscular, or cardiovascular disease; (5) non-smoker; (6) without flu or cold one week before or during the evaluation; (7) no use of psychotropic medication or muscle relaxants; (8) not pregnant; and (9) forced vital capacity greater than 80% of predicted. [12] Exclusion criteria were (1) MIP or MEP below 80% of predicted values; (2) inability to understand and perform pulmonary function maneuvers; (3) missed one of the RMS tests; (4) nasal congestion; (5) refusal to participate in any stage of the study; and (6) poor data quality. The study was approved by the research ethics committee of the Federal University of Rio Grande do Norte (number 2,631,047) and conducted according to the Declaration of Helsinki. All participants were informed and signed the informed consent form.

### Procedures

The clinical history, anthropometric, and pulmonary function data of participants were collected before MRP tests (MIP, MEP, SNIP, and SNEP).

The research protocol consisted of two sequences of assessments, according to simple randomization of tests. For sequence 1, participants performed inspiratory tests (MIP + SNIP$_{1-20}$) followed by expiratory tests (MEP + SNEP$_{1-20}$). Between the sequences, 30-minute intervals were provide to subjects. Sequence 2 inverted the order of tests to expiratory tests (MEP + SNEP$_{1-20}$) followed by performed inspiratory tests (MIP + SNIP$_{1-20}$). Participants were instructed and trained to perform the maneuvers correctly. Twenty consecutive SNIP and SNEP maneuvers were performed (i.e., SNIP$_{1-20}$ and SNIP$_{1-20}$).

### Spirometry

Spirometry was performed using the KoKo DigiDoser$^®$ (Longmont–USA) with participants sitting comfortably in a chair with a back and armrest. A nose clip was used to prevent the air from escaping through the upper airways during the procedure. Technical procedures followed the American Thoracic Society/European Respiratory Society [14], and reference values were calculated according to the Brazilian population [15].

### MIP and MEP

MIP and MEP were measured using a digital device V.2.0 (NEPEB—LabCare /UFMG, Belo Horizonte—MG, Brazil) connected to a disposable mouthpiece with a small orifice (~ 2 mm) to prevent glottic closure and reduce facial muscle activity. Technical criteria were followed by the European Respiratory Society [3] and the Brazilian Society of Pulmonology and Tisiology [14]. Participants were instructed to perform three to five MIP and MEP maneuvers; three should be acceptable and at least two reproducible [16]. To be considered valid, the maximum value obtained after five tests could not be 10% greater than the three best maneuvers [17]. The results obtained were compared to reference values for the Brazilian population [18].

### SNIP and SNEP

SNIP was performed using a nasal plug attached to one of the nostrils and consisted of performing a maximum and rapid inspiratory effort (sniff). The following criteria were used for accepting SNIP maneuvers: highest peak pressure value without leakage, duration of the

inspiratory effort up to 500 ms, maneuvers performed from functional residual capacity, peak pressure held for less than 50 ms, and pressure waveform displaying smooth curves [6, 20]. Technical criteria were followed by the European Respiratory Society [3], and reference values were obtained according to Araújo et al. [9].

SNEP was performed using an inflatable face mask (dead space of approximately 150 ml) fixed to the participant through a headgear (Vital Signs, New Jersey, USA). The face mask had two orifices: one to dissipate the pressure generated during the test (2 mm) and another to allow participants to breathe freely before the test and between maneuvers (15 mm). The second orifice was connected to a one-way inspiratory valve during the tests. The test consisted of maximum and rapid expirations (nose-blowing) from functional residual capacity with the mouth closed. The larger orifice was manually occluded during each maneuver and reopened right after the test. Participants were instructed on how to perform each maneuver. Twenty maneuvers of each test were conducted with a 30-second interval in between, and two tests were conducted 15 minutes before the protocol for familiarization.

### Statistical analysis

The Shapiro-Wilk test verified data normality. Parametric data were presented as mean and standard deviation (SD) and nonparametric data as a median and interquartile range [25% - 75%]. Paired or unpaired t-test was

performed for parametric variables and the Mann-Whitney test for nonparametric variables. The reliability of SNIP and SNEP was estimated using the intraclass correlation coefficient (ICC) between the highest peak pressure and the first reproducible maneuver.

Test-retest reliability was estimated using a two-way mixed-effects, type single rater, and consistency model based on McGraw and Wong (1996) convention [19]. The formula used was:

$$\frac{MS_R - MS_E}{MS_R + (k-1)MS_E}$$

where: MSR = mean square for rows; MSW = mean square for residual sources of variance; MSE = mean square for error; MSC = mean square for columns; n = number of subjects; k = number of raters/measurements. The ICC was stratified into low ($< 0.5$), moderate (between 0.5 and 0.75), good (between 0.75 and 0.90), and excellent ($> 0.90$) [18]. The standard error of measurement (SEM = SD $\times \sqrt{[1-ICC]}$) and minimum detectable change (MDC 95% = SEM $\times 1.96 \times \sqrt{2}$) were also calculated [19–22]. Statistical analyses were performed using the Statistical Package for the Social Sciences software version 22.0 (IBM Corporation, Armonk, NY, USA) and GraphPad Prism 8 (GraphPad Software, La Jolla, California, USA). A significance level of 5% ($p < 0.05$) was used.

The sample size was calculated using the G*Power software version 3.1.9.2 (Heinrich Heine–Universität Düsseldorf) considering the correlation between SNEP and MEP as the main variable and obtained a coefficient of determination ($r^2$) of 0.70 between SNEP and MEP variables, $\alpha$ of 0.001, and power (1 - $\beta$) of 0.99 and an effect size of 0.85. The sample size of 14 individuals was estimated, and this number was tripled to cover possible losses, totaling the final sample of 42 subjects.

## Results

The sample size of 14 individuals was estimated, and this number was tripled to cover possible losses, totaling the final sample of 42 subjects. Forty-nine participants were recruited, and 17 were excluded. From the exclusion, eight subjects (n = 8) were excluded due to poor pressure

signal data quality), one due (n = 1) due low MEP %, six (n = 6) due low SNIP % and one (n = 1) due the value of SNEP time curve up than 500 milliseconds. The final sample consisted of 32 participants (16 men and 16 women) with a median age of 23 years [ranging from 22 to 24]. Fig 1 represents the study flowchart. Anthropometric and spirometric data are presented in Table 1. Mean weight, height, and body mass index were 62 ± 7.4 kg, 1.68 ± 0.07 m, and 21.8 ± 1.7 kg/m$^2$, respectively. The data from age and FVC (%pred.) are considered nonparametric and expressed as median and interquartile range [25% - 75%]. The ICC,SEM and MDC for MIP and MEP was included on supplementary material.

Tables 2 and 3 present the absolute values of the highest and first reproducible maneuvers of SNIPs and SNEP. Relative and absolute frequencies showed that 60% (12 maneuvers) of the highest SNIP values and 55% (11 maneuvers) of the first reproducible occurred among the 11th and 20th maneuvers in men. For SNEP, these values were 55% and 40%, respectively. In women, 60% of the highest SNIP and 55% of the first reproducible maneuvers were also among the 11th and 20th maneuvers. SNEP measurements showed that 50% of the highest peak pressures and 60% of the first reproducible maneuvers were among the 11th and 20th maneuvers.

Table 4 presents the reliability among the highest peak pressures and first reproducible maneuvers, stratified by sex. We found excellent reliability of SNIP for men and women (ICC = 0.992); the total sample also showed excellent reliability (ICC = 0.994). The reliability of SNEP was good in men (ICC = 0.877) and excellent in women (ICC = 0.957). The total sample presented an ICC of 0.950. SEM and MDC were low in both tests.

Table 5 shows the coefficient of variation (CV) of the variables SNIP, SNEP, MIP and MEP, stratified by sex. Showing the total sample in all variables moderate CV, less than 30%.

## Discussion

The reliability of SNIP in adults' young healthy subjects was high in all contexts, ICC was close to the maximum, and SEM and MDC values were low. SNIP was considered reproducible according to our results. Moreover, SNIP is a widely used test [3, 10, 23], that is recommended to assess individuals with respiratory diseases (e.g., chronic obstructive pulmonary disease and neuromuscular diseases) [24, 26]. Other studies confirmed the good reliability of SNIP for assessing healthy adults [25–27] and children aged 8 to 11 years [28].

The reliability of SNEP in young healthy subjects was also good in men and excellent in women. SNEP is a novel method, and its reliability has never been studied. Morgan et al. [10] were the first to describe a nasal expiratory force in individuals with amyotrophic lateral sclerosis. In 2015, Ichikawa et al. [11] described the SNEP and showed a moderate correlation between SNEP and MEP; however, they did not assess the reliability or replicability. Unlike these authors, we demonstrated good reliability in men (ICC = 0.877) and excellent in women (ICC = 0.957); SEM and MDC were low and acceptable.

The ideal number of SNIP maneuvers to assess RMS is not a consensus. Lofaso et al. [7] considered that the learning effect could increase SNIP values after ten consecutive maneuvers in children and adults with different neuromuscular and respiratory diseases. The authors observed that the highest SNIP values among the first ten maneuvers were lower than the values of subsequent attempts and concluded that a relevant learning effect occurred after the 10th maneuver. As the learning effect may influence results, we performed 20 maneuvers in each test to obtain acceptable results. This is important mainly for SNEP, which is scarce in the literature and still under methodological development.

Although most studies presented ten maneuvers as sufficient for obtaining satisfactory results, previous studies predicted that more maneuvers could be required [7]. Marcelino et al.

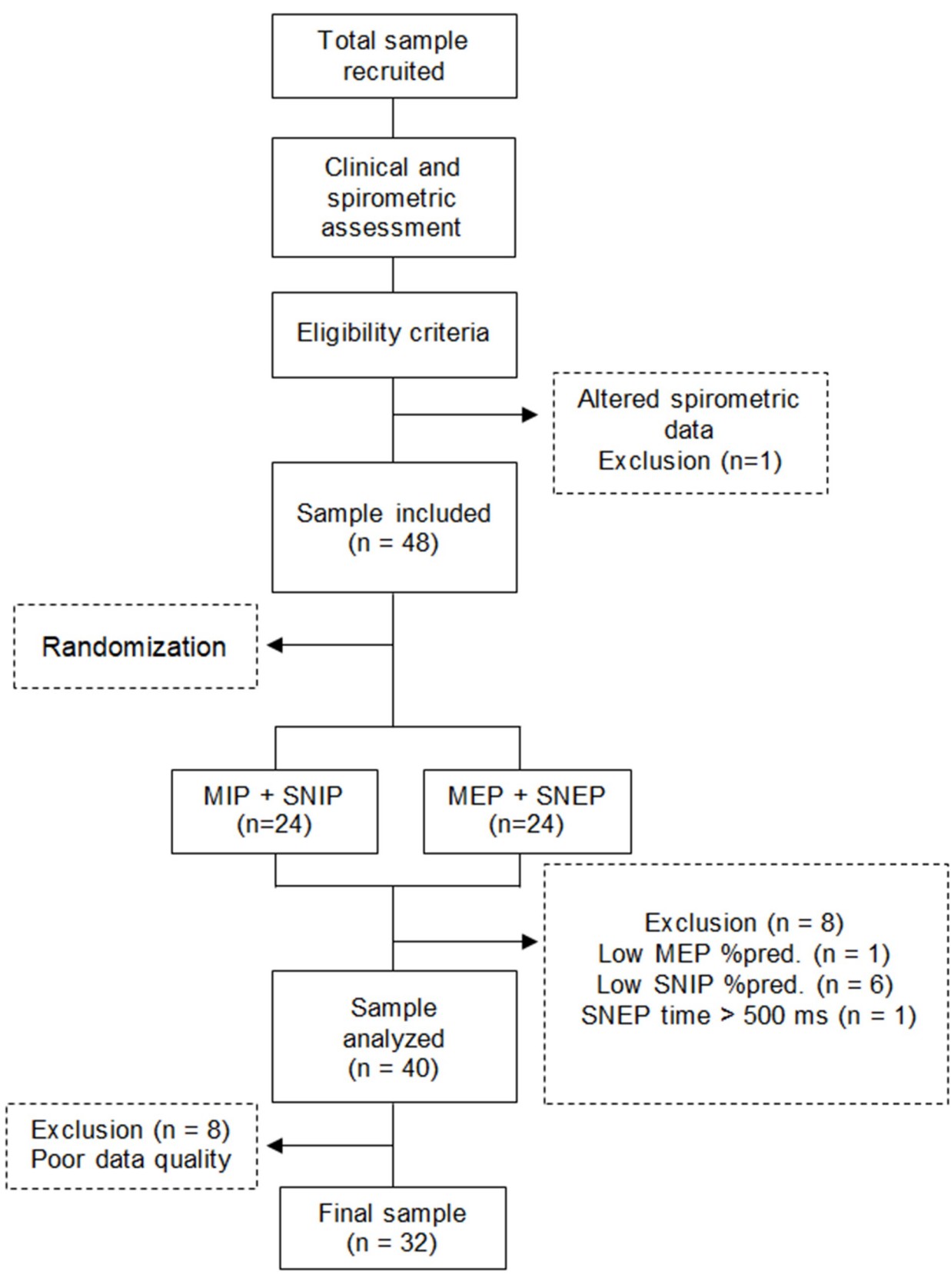

**Fig 1. Study flowchart.**

**Table 1. Anthropometric and spirometric data.**

| | Men (n = 16) | Women (n = 16) | Total |
|---|---|---|---|
| Age (years) | 23.3 [22–24] | 23.1 [21–24.7] | 23.1 [22–24] |
| Weight (kg) | 65.8 ± 7.1 | 58.1 ± 5.6* | 62 ± 7.4 |
| Height (m) | 1.71 ± 0.07 | 1.65 ± 0.05* | 1.68 ± 0.07 |
| BMI (kg/m$^2$) | 22.4 ± 1.7 | 21.2 ± 1.4* | 21.8 ± 1.7 |
| FVC (L) | 4.8 ± 0.5 | 3.9 ± 0.4* | 4.4 ± 0.6 |
| FVC (%pred.) | 90.3 [84.6–97.6] | 94.7 [88.1–109.2] * | 93.4 [85.5–101.8] |
| FEV$_1$ (L) | 4.05 ± 0.4 | 3.10 ± 0.3* | 3.5 ± 0.6 |
| FEV$_1$ (%pred.) | 90.0 ± 8.8 | 92.0 ± 10.6 | 91.0 ± 9.6 |
| FEV$_1$/ FVC | 0.84 ± 0.05 | 0.84 ± 0.08 | 0.84 ± 0.07 |
| MIP (cmH$_2$O) | 121.0 ± 21.6 | 82.2 ± 19.6 | 101.9 ± 21.4 |
| MEP (cmH$_2$O) | 128.6 ± 20.1 | 100.8 ± 36.7 | 115.4 ± 28.7 |
| MIP % | 88.6 ± 16 | 82.8 ± 19.2 | 114.3 ± 19.1 |
| MEP % | 87.8 ± 13.8 | 99.1 ± 32.3 | 94.7 ± 26.2 |

Parametric data were presented as mean and standard deviation and nonparametric data as a median and interquartile range [25% - 75%]. Inferential analysis was performed using an unpaired t-test or Mann-Whitney test. *p < 0.05; n—sample size; BMI—body mass index; FEV$_1$—forced expiratory volume in the first second; FVC —forced vital capacity; MIP—maximum inspiratory pressure; MEP—maximum expiratory pressure; kg—kilograms; m–meters; kg/m$^2$ –kilograms by squared meter; L–liters; %pred–percentage of predicted; cmH$_2$O –centimeters of water; %—percentage.

[28] concluded that twelve SNIP maneuvers were enough to obtain the highest peak pressure value in children up to 11 years old. They also showed good reliability and reported that 42% of the highest SNIP values were among the 6th and 10th maneuvers; 3% of participants achieved the same performance over 12 maneuvers. Corroborating Lofaso et al. [7] and Marcelino et al. [28], we found that 75% (24 maneuvers) of the highest SNIP values and 69% (22

**Table 2. Highest SNIP maneuvers and first reproducible maneuver.**

| Sample | SNIP (n° maneuver) | 1st reproducible (n° maneuver) | Sample | SNIP (n° maneuver) | 1st reproducible (n° maneuver) |
|---|---|---|---|---|---|
| Men | | | Women | | |
| 1 | 149 (07) | 141 (09) | 17 | 86 (13) | 85 (18) |
| 2 | 157 (07) | 145 (10) | 18 | 105 (18) | 102 (14) |
| 3 | 133 (20) | 130 (13) | 19 | 79 (04) | 78 (07) |
| 4 | 141 (09) | 138 (12) | 20 | 77 (10) | 76 (19) |
| 5 | 129 (11) | 125 (02) | 21 | 102 (08) | 97 (05) |
| 6 | 138 (20) | 136 (18) | 22 | 74 (20) | 69 (16) |
| 7 | 93 (14) | 84 (20) | 23 | 108 (17) | 101 (20) |
| 8 | 122 (19) | 117 (18) | 24 | 92 (19) | 86 (15) |
| 9 | 131 (10) | 130 (20) | 25 | 89 (17) | 88 (12) |
| 10 | 84 (20) | 77 (18) | 26 | 123 (19) | 122 (18) |
| 11 | 143 (17) | 135 (18) | 27 | 82 (18) | 79 (20) |
| 12 | 119 (20) | 103 (19) | 28 | 93 (01) | 89 (04) |
| 13 | 91 (19) | 87 (10) | 29 | 95 (19) | 95 (17) |
| 14 | 89 (16) | 84 (15) | 30 | 93 (20) | 89 (15) |
| 15 | 105 (17) | 105 (12) | 31 | 101 (12) | 94 (10) |
| 16 | 135 (16) | 129 (10) | 32 | 102 (13) | 99 (10) |

The absolute pressure in centimeters of water (cmH$_2$O); n°—Number of the maneuver are in parentheses.

**Table 3. Highest SNEP and first reproducible maneuver.**

| Sample | SNEP (n° maneuver) | 1st reproducible (n° maneuver) | Sample | SNEP (n° maneuver) | 1st reproducible (n° maneuver) |
|---|---|---|---|---|---|
| Men | | | Women | | |
| 1 | 137 (12) | 130 (13) | 17 | 127 (15) | 121 (16) |
| 2 | 173 (18) | 140 (19) | 18 | 148 (14) | 147 (15) |
| 3 | 119 (09) | 118 (10) | 19 | 92 (02) | 91 (07) |
| 4 | 158 (18) | 147 (19) | 20 | 94 (15) | 93 (14) |
| 5 | 155 (19) | 117 (04) | 21 | 76 (10) | 68 (07) |
| 6 | 90 (04) | 88 (07) | 22 | 87 (07) | 79 (19) |
| 7 | 153 (08) | 123 (09) | 23 | 83 (12) | 76 (17) |
| 8 | 122 (18) | 114 (11) | 24 | 131 (18) | 126 (08) |
| 9 | 138 (13) | 119 (07) | 25 | 92 (02) | 79 (16) |
| 10 | 116 (12) | 116 (08) | 26 | 128 (20) | 125 (19) |
| 11 | 105 (16) | 102 (15) | 27 | 66 (11) | 21 (13) |
| 12 | 144(15) | 144 (16) | 28 | 94 (10) | 89 (18) |
| 13 | 127 (07) | 106 (08) | 29 | 115 (19) | 112 (18) |
| 14 | 108 (11) | 102 (20) | 30 | 99 (13) | 98 (17) |
| 15 | 131 (14) | 128 (12) | 31 | 79 (20) | 77 (07) |
| 16 | 129 (09) | 127 (02) | 32 | 102 (10) | 99 (19) |

The absolute pressure in centimeters of water ($cmH_2O$); n°—Number of the maneuver are in parentheses.

maneuvers) of the reproducible maneuvers were obtained in the last ten maneuvers. Also, 65% of the highest SNEP values and 62% of the reproducible maneuvers were achieved among the 11th and 20th maneuvers. According to the results, respiratory muscle fatigue did not influence the tests.

Study limitations included the small sample and a possible risk of bias (assessments and data analyses were not blind). The sample used is from healthy adults so thei results cannot be extrapolated directly to the elderly population or people with neuromuscular involvement, however it serves as a precedent for future clinical investigations. Moreover, the sample size was estimated without gender differentiation, and outcome assessments and data analyses were not blinded. Therefore, results must be interpreted with caution, and future studies considering these limitations are needed.

**Table 4. Reliability of SNIP, SNEP, MIP and MEP between the highest peak and first reproducible maneuvers.**

| | Men (n = 16) | Women (n = 16) | Total (n = 32) |
|---|---|---|---|
| | SNIP | | |
| ICC [95% CI] | 0.992 [0.977–0.997] | 0.992 [0.977–0.997] | 0.994 [0.988–0.997] |
| SEM ($cmH_2O$) | 2.073 | 1.152 | 1.820 |
| MDC ($cmH_2O$) | 5.747 | 3.193 | 5.043 |
| | SNEP | | |
| ICC [95%CI] | 0.877 [0.648–0.957] | 0.957 [0.878–0.985] | 0.950 [0.997–0.976] |
| SEM ($cmH_2O$) | 7.660 | 6.115 | 6.031 |
| MDC ($cmH_2O$) | 21.231 | 16.950 | 16.716 |

n–Sample size; ICC—Intraclass correlation coefficient; SEM—Standard error of measurement; MDC—Minimum detectable change; 95% CI—Confidence interval; $cmH_2O$ - Centimeters of water

**Table 5. Coefficient of variation of SNIP, SNEP, MIP and MEP values.**

|  | Men (n = 16) | Women (n = 16) | Total (n = 32) |
|---|---|---|---|
| SNIP | | | |
| Mean (cmH$_2$O) | 121.6 | 88.3 | 103.7 |
| SD (cmH$_2$O) | 23.2 | 12.9 | 23.3 |
| CV (%) | 19.1 | 14.6 | 22.5 |
| SNEP | | | |
| Mean (cmH$_2$O) | 131.6 | 94.9 | 111.5 |
| SD (cmH$_2$O) | 21.8 | 22.8 | 26.5 |
| CV (%) | 16.6 | 24 | 23.8 |
| MIP | | | |
| Mean (cmH$_2$O) | 121 | 82.2 | 101.9 |
| SD (cmH$_2$O) | 21.7 | 19.6 | 25.4 |
| CV (%) | 17.9 | 23.8 | 24.9 |
| MEP | | | |
| Mean (cmH$_2$O) | 128.6 | 100.8 | 115.4 |
| SD (cmH$_2$O) | 20.2 | 33.7 | 28.7 |
| CV (%) | 15.7 | 33.4 | 24.9 |

SD–mean and standard deviation; CV = coefficient of variation; cmH$_2$O - Centimeters of water; %—percentage

## Conclusion

The ICC, SEM, and MDC were high for SNIP and SNEP, indicating their reliability in healthy young adults. The SNIP and SNEP maneuvers are simple and inexpensive, easy to perform, and should be used as complementary to MIP and MEP to improve the diagnosis and monitoring of muscle. In conclusion, following our results we suggest that twenty maneuvers were required to obtain the best results when the test is used for diagnosis and monitoring of muscle weakness in several diseases in which a deficit in respiratory muscle strength is related to the natural history of the disease.

## Supporting information

**S1 Table. Reliability of MIP and MEP between the highest peak and first reproducible maneuvers.** This is the S1 Table legend n–Sample size; ICC—Intraclass correlation coefficient; SEM—Standard error of measurement; MDC—Minimum detectable change; 95% CI—Confidence interval; cmH$_2$O - Centimeters of water.
(DOCX)

**S1 Data. Data froam de Study patients.** This is the S1 Fig legend. No legend.
(XLSX)

## Acknowledgments

"The authors thank Provatis Academic Services for providing scientific language translation, revision, and editing."

## Author Contributions

**Conceptualization:** Jackson C. C de Lima, Vanessa R. Resqueti, Guilherme A. F. Fregonezi.

**Data curation:** Vanessa R. Resqueti, Ana Aline Marcelino, Jéssica Danielle M. da Fonsêca, Fernando A. Lavezzo Dias, Matias Otto-Yañez, Guilherme A. F. Fregonezi.

**Formal analysis:** Jackson C. C de Lima, Vanessa R. Resqueti, Ana Aline Marcelino, Jéssica Danielle M. da Fonsêca, Ana Lista Paz, Fernando A. Lavezzo Dias, Matias Otto-Yañez, Guilherme A. F. Fregonezi.

**Funding acquisition:** Vanessa R. Resqueti, Guilherme A. F. Fregonezi.

**Investigation:** Jackson C. C de Lima, Vanessa R. Resqueti.

**Methodology:** Jackson C. C de Lima, Vanessa R. Resqueti, Ana Aline Marcelino, Jéssica Danielle M. da Fonsêca, Matias Otto-Yañez, Guilherme A. F. Fregonezi.

**Project administration:** Jackson C. C de Lima, Vanessa R. Resqueti.

**Resources:** Vanessa R. Resqueti, Jéssica Danielle M. da Fonsêca, Fernando A. Lavezzo Dias.

**Supervision:** Vanessa R. Resqueti, Guilherme A. F. Fregonezi.

**Validation:** Jackson C. C de Lima, Vanessa R. Resqueti, Ana Aline Marcelino, Matias Otto-Yañez, Guilherme A. F. Fregonezi.

**Visualization:** Jackson C. C de Lima, Vanessa R. Resqueti, Jéssica Danielle M. da Fonsêca, Ana Lista Paz, Guilherme A. F. Fregonezi.

**Writing – original draft:** Jackson C. C de Lima, Vanessa R. Resqueti, Ana Aline Marcelino, Jéssica Danielle M. da Fonsêca, Ana Lista Paz, Matias Otto-Yañez, Guilherme A. F. Fregonezi.

**Writing – review & editing:** Jackson C. C de Lima, Vanessa R. Resqueti, Ana Lista Paz, Fernando A. Lavezzo Dias, Matias Otto-Yañez, Guilherme A. F. Fregonezi.

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
