## [Decision Letter · Decision Letter 0]

29 Aug 2022

PONE-D-22-15898Reliability of maximal respiratory nasal pressure tests in healthy young adultsPLOS ONE

Dear Dr. Fregonezi,

Thank you for submitting your manuscript to PLOS ONE. After careful consideration, we feel that it has merit but does not fully meet PLOS ONE’s publication criteria as it currently stands. Therefore, we invite you to submit a revised version of the manuscript that addresses the points raised during the review process.

ACADEMIC EDITOR:Dear Authors,two experts in the field reviewed your manuscript and reported several major methodological issues you should consider while revising the paper.

We look forward to receiving your revised manuscript.

Kind regards,

Emiliano Cè

Academic Editor

PLOS ONE

https://journals.plos.org/plosone/s/file?id=ba62/PLOSOne_formatting_sample_title_authors_affiliations.pdf".

a) Did participants provide their written or verbal informed consent to participate in this study?

Reviewers' comments:

Reviewer's Responses to Questions

**Comments to the Author**

1. Is the manuscript technically sound, and do the data support the conclusions?

Reviewer #1: Partly

Reviewer #2: Yes

2. Has the statistical analysis been performed appropriately and rigorously? 

Reviewer #1: No

Reviewer #2: Yes

3. Have the authors made all data underlying the findings in their manuscript fully available?

Reviewer #1: No

Reviewer #2: Yes

4. Is the manuscript presented in an intelligible fashion and written in standard English?

Reviewer #1: Yes

Reviewer #2: Yes

5. Review Comments to the Author

Reviewer #1: The scope of the review is interesting, however I have made some remarks regarding this study.

Introduction

Please provide the better rationale for need to research the reliability of the SNAP and SNIP in the context of the existing knowledge on the complementarity of SNIP to MIP in assessment of RMS. The MIP and MEP are reliable and widely used due to their simplicity, practicality, and tolerability. What is common and what differ the methods (SNIP vs. MIP or SNAP vs. MEP).

Sample size and statistic

Did you estimated the total sample size or with differentiation of the gender. The actual and predicted value of MEP in men is different to value in women population.

Did you take into account the potential risk of losing participants in recruiting?

Describe what model and formula of intraclass correlation coefficient (ICC) did you used?

Did were calculated coefficient of variation? Please, provide coefficient of variation. To results section.

I have propose to assess the criterion validity, assuming the MEP, MIP as a gold standard assessment methods.

Please extend the tables with the ICC results for 10 to 20 repetitions to see the reliability of the measurements for 10, 11 to 20 repetitions.

Conclusions

Present the practical implication of your results.

You wrote: „Twenty maneuvers were required to obtain the best results without interfering with the performance of each test”. I can’t agree with it, because the results are show only as a mean of 20 repetition.

The limitations of this study are not discussed.

Reviewer #2: This manuscript sought to investigate the reliability and sensitivity of SNIP and SNEP as possible index to evaluate respiratory muscles strength. High to very-high ICC with low SEM% and MDC95% were found for both indexes in both sexes.

The study is on the whole well written and as some potentialities. Introduction is concise and the scope clearly stated. I would only stress better the importance to differentiate the results between women and men. Although I could understand the rationale, I think it would be better specifying it better.

Methods are clearly reported. Just a suggestion: since you are proposing SNIP and SNEP as complementary indexes to assess respiratory muscle force, I would also show the correlations between SNIP and MIP and SNEP and MEP. Moreover, in the statistical analysis section, please provide some more details on the ICC approach used.

Results are clearly reported. Tables are clear.

Discussion: I would strongly stress here that the data are (although encouraging) from a sample of healthy and young participants. This reduce the generatability of the results to an older population or people with neurodegenerative diseases.

6. PLOS authors have the option to publish the peer review history of their article (what does this mean?). If published, this will include your full peer review and any attached files.

Reviewer #1: No

Reviewer #2: No

---

## [Author Response · Author response to Decision Letter 0]

1 Mar 2023

Dear Prof. Dr. Emiliano Cè

Academic Editor

PLOS ONE

Natal, February 2022 

Subject: Revision and resubmission of Manuscript PONE-D-22-15898

Dear Prof. Dr. Emiliano Cè

Thank you for revising our manuscript. We appreciate the reviewers’ complimentary comments and suggestions. We have revised the manuscript following the recommendations.

Please find attached a point-by-point response to the reviewer’s comments. We hope that you find our answers satisfactory and that the manuscript is now acceptable for publication.

Sincerely,

Prof. Dr. Guilherme Fregonezi

Reviewer #1

Reviewer #1: The scope of the review is interesting; however, I have made some remarks regarding this study.

Introduction

Please provide a better rationale for need to research the reliability of the SNEP and SNIP in the context of the existing knowledge on the complementarity of SNIP to MIP in the assessment of RMS. The MIP and MEP are reliable and widely used due to their simplicity, practicality, and tolerability. What is common and what differs between the methods (SNIP vs. MIP or SNAP vs. MEP)?

Authors: Thank you. The rationale was improved. New information and new references were added. Bellow the text.

“Regarding the assessment of expiratory muscle strength, Morgan et al. [10] described for the first time a method similar to SNIP, the sniff nasal expiratory force. Subsequently, Ichikawa et al. [11] presented the assessment of sniff nasal expiratory pressure (SNEP). Since the technical development, the MIP and MEP tests differ from that of SNIP and SNEP from the air inlet route, the lung volume where the maneuver is carried out, the duration of the test, and possible muscle activation, therefore reliability cannot be extrapolated between tests. Furthermore, gender could influence SNIP and SNEP results as well as it influences other respiratory variables. Preliminary data, from SNIP and MIP comparison, suggested that both tests should be used for early detection of respiratory muscle weakness to avoid over/under diagnoses [12,13]. The focus on precision rehabilitation requires that tests for evaluating respiratory muscle function be reliable and reproducible with each other. Tests that have low reliability and reproducibility cannot be presented as an evaluation alternative. This is why it is necessary to establish these values for these evaluations. Nevertheless, the reliability and reproducibility of SNIP and SNEP are still unknown. Therefore, this study aimed to analyze the reproducibility and reliability of SNIP and SNEP in healthy young adults.” 

Reviewer #1: Sample size and statistic. Did you estimate the total sample size or with differentiation of the gender? The actual and predicted value of MEP in men is different from to value in the women population.

Authors: Thank you for the observation. No gender differentiation was performed during sample size calculation. Gender is a relevant information for respiratory muscle strength; however, given the scarcity of research on gender differences, we opted to calculate all groups together. Due to the relevance of this information, we included it as a limitation of the study.

“Study limitations included the small sample and a possible risk of bias (assessments and data analyses were not blind). The sample used is healthy adults, so their results cannot be directly extrapolated to the elderly population or people with neuromuscular involvement, but it is useful to leave a baseline precedent. Moreover, sample size was estimated without gender differentiation, and outcome assessments and data analyses were not blinded. Therefore, results must be interpreted with caution, and future studies considering these limitations are needed.”

Reviewer #1: Did you take into account the potential risk of losing participants in recruiting?

Authors: Thank you for the note. We included additional information regarding sample size calculation. Bellow is the new text with additional information.

 “Sample size was calculated using the G*Power software version 3.1.9.2 (Heinrich Heine – Universität Düsseldorf) considering the correlation between SNEP and MEP as the main variable and obtained using a coefficient of determination (r2) of 0.70 between SNEP and MEP variables, α of 0.001, and power (1 - β) of 0.99 and an effect size of 0.85. The sample size of 14 individuals was estimated, and this number was tripled to cover possible losses, totaling the final sample of 42 subjects.” 

Reviewer #1: Describe what model and formula of intraclass correlation coefficient (ICC) you used.

Authors: Thank you for the comment. We have added information about the model used in the statistical analysis section of the manuscript.

“Test-retest reliability was estimated using a two-way mixed-effects, type single rater and consistency model described by McGraw and Wong (1996) convention. The formula used was:

where: MSR = mean square for rows; MSW = mean square for residual sources of variance; MSE = mean square for error; MSC = mean square for columns; n = the number of subjects; k = the number of raters/measurements. The ICC was stratified into low (< 0.5), moderate (between 0.5 and 0.75), good (between 0.75 and 0.90), and excellent (> 0.90)”

Reviewer #1: Did has calculated the coefficient of variation? Please, provide the coefficient of variation. To results section. I have proposed to assess the criterion validity, assuming the MEP, and MIP as a gold standard assessment method.

Authors: We did not calculate the coefficient of variation because the objective of the study was not analysis dispersion. The use of MIP and MEP to assess the criterion validity also was not the objective of the study. Thank you for the suggestion, we will take it into account in the next studies.

Reviewer #1: Please extend the tables with the ICC results for 10 to 20 repetitions to see the reliability of the measurements for 10, 11 to 20 repetitions.

Authors: Thank you for the suggestion. I am not sure we quite understand your suggestion. Table 4 showed the results for the aim os the study which was to analyze the reliability of SNIP and SNEP using the intraclass correlation coefficient (ICC) between the highest peak pressure and the first reproducible maneuver. In table 2 and table 3 is possible to observe the order of the highest maneuver and the first reproducible maneuver. 

The results of Tables 2, 3, and 4 were described: 

Regarding Tables 2 and 3

“The absolute values of the highest and first reproducible maneuvers of SNIPs and SNEP. Relative and absolute frequencies showed that 60% (12 maneuvers) of the highest SNIP values and 55% (11 maneuvers) of the first reproducible occurred among the 11th and 20th maneuvers in men. For SNEP, these values were 55% and 40%, respectively. In women, 60% of the highest SNIP and 55% of the first reproducible maneuvers were also among the 11th and 20th maneuvers. SNEP measurements showed that 50% of the highest peak pressures and 60% of the first reproducible maneuvers were among the 11th and 20th maneuvers.”

Regarding Table 4

“Table 4 presents the reliability among the highest peak pressures and first reproducible maneuvers stratified by sex. Table 4 shows the excellent reliability of SNIP for men and women (ICC = 0.992); the total sample also showed excellent reliability (ICC = 0.994). The reliability of SNEP was good in men (ICC = 0.877) and excellent in women (ICC = 0.957). The total sample presented an ICC of 0.950. SEM and MDC were low in both tests.”

Reviewer #1: Conclusions. Present the practical implication of your results.

Authors: Thank you for the suggestion. We include the practical implications of our results in the conclusion. 

The ICC, SEM, and MDC were high for SNIP and SNEP, indicating their reliability in healthy young adults. The SNIP and SNEP maneuvers are simple and inexpensive, easy to perform, and should be used as complementary to MIP and MEP to improve the diagnosis and monitoring of muscle. In conclusion, following our results we suggest that twenty maneuvers were required to obtain the best results when the test is used for diagnosis and monitoring of muscle weakness in several diseases in which a deficit in respiratory muscle strength is related to the natural history of the disease.

Reviewer #1: You wrote: „Twenty maneuvers were required to obtain the best results without interfering with the performance of each test”. I can’t agree with it, because the results are shown only as a mean of 20 repetitions.

Authors: Thank you for your observation. The conclusion was based on our results: 

“We found that 75% (24 maneuvers) of the highest SNIP values and 69% (22 maneuvers) of the reproducible maneuvers were obtained in the last ten maneuvers. Also, 65% of the highest SNEP values and 62% of the reproducible maneuvers were achieved among the 11th and 20th maneuvers. According to results, respiratory muscle fatigue did not influence the tests.” 

We add new information to the conclusion to express our results.

“The ICC, SEM, and MDC were high for SNIP and SNEP, indicating their reliability in healthy young adults. The SNIP and SNEP maneuvers are simple and inexpensive, easy to perform, and should be used as complementary to MIP and MEP to improve the diagnosis and monitoring of muscle weakness. In conclusion, following our results we suggest that twenty maneuvers were required to obtain the best results when the test is used diagnosis and monitoring of muscle weakness in several diseases in which a deficit in respiratory muscle strength is related to the natural history of the disease.” 

Reviewer #1: The limitations of this study are not discussed.

Authors: Thank you for your suggestion. We have improved the text of the study's limitations. We hope that it covers all the limitations pointed out in the review.

“ Study limitations included the small sample and a possible risk of bias (assessments and data analyses were not blind). The sample used is healthy adults, so their results cannot be directly extrapolated to the elderly population or people with neuromuscular involvement, but it is useful to leave a baseline precedent. Moreover, sample size was estimated without gender differentiation, and outcome assessments and data analyses were not blinded. Therefore, results must be interpreted with caution, and future studies considering these limitations are needed.”

Reviewer #2

This manuscript sought to investigate the reliability and sensitivity of SNIP and SNEP as the possible index to evaluate respiratory muscle strength. High to very-high ICC with low SEM% and MDC95% were found for both indexes in both sexes. The study is on the whole well written and has some potential.

The introduction is concise and the scope is clearly stated. I would only stress better the importance to differentiate the results between women and men. Although I could understand the rationale, I think it would be better to specify it better.

Authors: Thank you for your suggestion. We include a phrase on the rationale to express the importance of gender. 

“ Regarding the assessment of expiratory muscle strength, Morgan et al. [10] described for the first time a method similar to SNIP, the sniff nasal expiratory force. Subsequently, Ichikawa et al. [11] presented the assessment of sniff nasal expiratory pressure (SNEP). Since the technical development, the MIP and MEP tests differ from that of SNIP and SNEP from the air inlet route, the lung volume where the maneuver is carried out, the duration of the test, and possible muscle activation, therefore reliability cannot be extrapolated between tests. Furthermore, gender could influence SNIP and SNEP results as well as other respiratory variables. Preliminary data, from SNIP and MIP comparison, suggested that both tests should be used for early detection of respiratory muscle weakness to avoid over/under diagnoses [12,13]. The focus on precision rehabilitation requires that tests for evaluating respiratory muscle function be reliable and reproducible with each other. Tests that have low reliability and reproducibility cannot be presented as an evaluation alternative. This is why it is necessary to establish these values for these evaluations. Nevertheless, the reliability and reproducibility of SNIP and SNEP are still unknown. Therefore, this study aimed to analyze the reproducibility and reliability of SNIP and SNEP in healthy young adults.” 

Reviewer #2: Methods are reported. Just a suggestion: since you are proposing SNIP and SNEP as complementary indexes to assess respiratory muscle force, I would also show the correlations between SNIP and MIP and SNEP and MEP. Moreover, in the statistical analysis section, please provide some more details on the ICC approach used.

Authors: Thank you for the suggestion. The data from correlation will be included in another manuscript that is now in preparation with greater sample size. We have added information about the model used in the statistical analysis section of the manuscript.

“Test-retest reliability was estimated using a two-way mixed-effects, type single rater and consistency model described by McGraw and Wong (1996) convention. The formula used was:

Where: MSR = mean square for rows; MSW = mean square for residual sources of variance; MSE = mean square for error; MSC = mean square for columns; n = the number of subjects; k = the number of raters/measurements. The ICC was stratified into low (< 0.5), moderate (between 0.5 and 0.75), good (between 0.75 and 0.90), and excellent (> 0.90)”

Reviewer #2: Results are reported. The tables are clear.

Authors: Thank you.

Reviewer #2: Discussion: I would strongly stress here that the data are (although encouraging) from a sample of healthy and young participants. This reduces the generatability of the results to an older population or people with neurodegenerative diseases.

Authors: Thank you. We stress this information in the first paraph of the discussion and, is included in the study limitation text.

“The reliability of SNIP in young healthy subjects was high in all contexts, ICC was close to the maximum, and SEM and MDC values were low. SNIP is reproducible, widely used [3,10,23], and recommended to assess individuals with respiratory diseases (e.g., chronic obstructive pulmonary disease and neuromuscular diseases) [24, 26]. Other studies confirmed the good reliability of SNIP for assessing healthy adults [25, 26] and children aged 8 to 11 years [28].”

“The reliability of SNEP in young healthy subjects was also good in men and excellent in women. SNEP is a novel method, and its reliability has never been studied. Morgan et al. [10] were the first to describe a nasal expiratory force in individuals with amyotrophic lateral sclerosis. In 2015, Ichikawa et al. [11] described the SNEP and showed a moderate correlation between SNEP and MEP; however, they did not assess the reliability or replicability. Unlike these authors, we demonstrated good reliability in men (ICC = 0.877) and excellent in women (ICC = 0.957); SEM and MDC were low and acceptable.”

“Study limitations included the small sample and a possible risk of bias (assessments and data analyses were not blind). The sample used is from healthy adults so the results cannot be extrapolated directly to the elderly population or people with neuromuscular involvement, however it serves as a precedent for future clinical investigations. Moreover, the sample size was estimated without gender differentiation and assessments, and data analyses were not blinded. Therefore, results must be interpreted with caution, and future studies considering these limitations are needed.”

---

## [Decision Letter · Decision Letter 1]

27 Mar 2023

PONE-D-22-15898R1Reliability of maximal respiratory nasal pressure tests in healthy young adultsPLOS ONE

Dear Dr. Augusto de Freitas Fregonezi,

Thank you for submitting your manuscript to PLOS ONE. After careful consideration, we feel that it has merit but does not fully meet PLOS ONE’s publication criteria as it currently stands. Therefore, we invite you to submit a revised version of the manuscript that addresses the points raised during the review process. Dear Authors, one expert in the field re-reviewed your manuscript founding several major points you should consider during the revision process.

We look forward to receiving your revised manuscript.

Kind regards,

Emiliano Cè

Academic Editor

PLOS ONE

Reviewers' comments:

Reviewer's Responses to Questions

**Comments to the Author**

1. If the authors have adequately addressed your comments raised in a previous round of review and you feel that this manuscript is now acceptable for publication, you may indicate that here to bypass the “Comments to the Author” section, enter your conflict of interest statement in the “Confidential to Editor” section, and submit your "Accept" recommendation.

Reviewer #1: All comments have been addressed

2. Is the manuscript technically sound, and do the data support the conclusions?

Reviewer #1: Partly

3. Has the statistical analysis been performed appropriately and rigorously? 

Reviewer #1: No

4. Have the authors made all data underlying the findings in their manuscript fully available?

Reviewer #1: Yes

5. Is the manuscript presented in an intelligible fashion and written in standard English?

Reviewer #1: Yes

6. Review Comments to the Author

Reviewer #1: For reviewer are some issues not clear describe and required clarification:

1. Procedures

The autors wrote: „The research protocol consisted of two sequences of assessments, according to simple randomization of tests. For sequence 1, participants performed inspiratory tests (MIP + SNIP1-20) followed by expiratory tests (MEP + SNEP1-20) after a 30-minute interval. Sequence 2 inverted the order of tests.”

Is there a 30-minute gap between the inspiration test and the exhalation test? Was this interval between sequence 1 and sequence 2?

2. The authors in response to the review wrote: „ The use of MIP and MEP to assess the criterion validity also was not the objective of the study”.

Please explain which outcomes allow to formulate that conclusion: „The SNIP and SNEP maneuvers are simple and inexpensive, easy to perform, and should be used as complementary to MIP and MEP to improve the diagnosis and monitoring of muscle.

3. In introduction the authors state: Nevertheless, the reliability and reproducibility of SNIP and SNEP are still unknown.

In discussion section the authors wrote: „SNIP is reproducible, widely used [3,10,23], and recommended to assess individuals with respiratory diseases (e.g., chronic obstructive pulmonary disease and neuromuscular diseases) [24, 26}”

4. Results:

Please explain, why 17 participants were excluded from the study.

5. Statistical analysis:

Which data were parametric and which data were nonparametric?

7. PLOS authors have the option to publish the peer review history of their article (what does this mean?). If published, this will include your full peer review and any attached files.

Reviewer #1: No

---

## [Author Response · Author response to Decision Letter 1]

15 May 2023

Dear Prof. Dr. Emiliano Cè

Academic Editor

PLOS ONE

Natal, may 2022 

Subject: Revision and resubmission of Manuscript PONE-D-22-15898

Dear Prof. Dr. Emiliano Cè

 Thank you for revising our manuscript. We appreciate the reviewer's complimentary comments and suggestions. We have revised the manuscript following the recommendations. Please find attached a point-by-point response to the reviewer’s comments. We hope that you find our answers satisfactory and that the manuscript is now acceptable for publication.

Sincerely,

Prof. Dr. Guilherme Fregonezi

 Reviewer #1

For reviewer are some issues not clear describe and required clarification:

 1. Procedures The author's wrote: The research protocol consisted of two sequences of assessments, according to simple randomization of tests. For sequence 1, participants performed inspiratory tests (MIP + SNIP1-20) followed by expiratory tests (MEP + SNEP1-20) after a 30-minute interval. Sequence 2 inverted the order of tests.” Is there a 30-minute gap between the inspiration test and the exhalation test? Was this interval between sequence 1 and sequence 2?

Authors: Thank you for your question. Yes, there was a gap between sequences. We add new information to the paragraph to improve the text. Bellow the paragraph as described in the text of the manuscript procedures section: 

Procedures 

The clinical history, anthropometric, and pulmonary function data of participants were collected before MRP tests (MIP, MEP, SNIP, and SNEP). The research protocol consisted of two sequences of assessments, according to simple randomization of tests. For sequence 1, participants performed inspiratory tests (MIP + SNIP1-20) followed by expiratory tests (MEP + SNEP1-20). Between the sequences, 30-minute intervals was provide to subjects. Sequence 2 inverted the order of tests to expiratory tests (MEP + SNEP1-20) followed by performed inspiratory tests (MIP + SNIP1-20). Participants were instructed and trained to perform the maneuvers correctly. Twenty consecutive SNIP and SNEP maneuvers were performed (i.e., SNIP1-20 and SNIP1-20).

2. The authors in response to the review wrote: “ The use of MIP and MEP to assess the criterion validity also was not the objective of the study”. Please explain which outcomes allow to formulate that conclusion:The SNIP and SNEP maneuvers are simple and inexpensive, easy to perform, and should be used as complementary to MIP and MEP to improve the diagnosis and monitoring of muscle.

Authors: Dear reviewer, thanks for the opportunity to change our manuscript. We made a mistake in the response to the review. We used the MIP and MEP as a gold standard to analyze the criterion validity of SNIP and SNEP. We include in the manuscript one new table, table 5, with information about the coefficient of variation of SNIP, SNEP, MIP, and MEP values. Moreover, we add data in supplementary material with information on ICC, SEM, and MDC to MIP and MEP. The conclusions were formulated based on the results of reliability analyses. Bellow the new table add to the results section of the manuscript. 

 Table 5 shows the coefficient of variation (CV) of the variables SNIP, SNEP, MIP and MEP, stratified by sex. Showing the total sample in all variables moderate CV, less than 30%.

Table 5. Table 5. Coefficient of variation of SNIP, SNEP, MIP and MEP values.

SD – mean and standard deviation; CV = coefficient of variation; cmH2O - Centimeters of water; % - percentage

 

3. In introduction the authors state: Nevertheless, the reliability and reproducibility of SNIP and SNEP are still unknown.

 In discussion section the authors wrote: „ SNIP is reproducible, widely used [3,10,23], and recommended to assess individuals with respiratory diseases (e.g., chronic obstructive pulmonary disease and neuromuscular diseases) [24, 26}”

Authors: Thanks for your note. We included the word adult in the introduction and discussion sections based on literature information. The phrase in the discussion section was modified during the process of correction and translation into English. We have included the word adult in the introduction and rephrased the sentence in the discussion section. Below is the final version: 

Introduction heading: 

Nevertheless, the reliability and reproducibility of SNIP and SNEP in adults are still unknown. 

Discussion heading:

The reliability of SNIP in adults’ young healthy subjects was high in all contexts, ICC was close to the maximum, and SEM and MDC values were low. SNIP was considered reproducible according to our results. Moreover, SNIP is a widely used test [3,10,23],  that is recommended to assess individuals with respiratory diseases (e.g., chronic obstructive pulmonary disease and neuromuscular diseases) [24, 26]. Other studies confirmed the good reliability of SNIP for assessing healthy adults [25, 26] and children aged 8 to 11 years [28].

4. Results: Please explain, why 17 participants were excluded from the study.

Authors: This information was included in Fig. 1 of the study flowchart and is now also included in the first paragraph of the results sections.

"The sample size of 14 individuals was estimated, and this number was tripled to cover possible losses, totaling the final sample of 42 subjects. Forty-nine participants were recruited, and 17 were excluded. From the exclusion, eight subjects  (n = 8) were excluded due to poor pressure signal  data quality), one due (n = 1) due low MEP %, six (n = 6) due low SNIP % and one (n = 1) due the value of SNEP time curve up than 500 millisecond "

 5. Statistical analysis: Which data were parametric and which data were nonparametric?

Authors:Thank you for your question. This information can be found in the footnotes of Table 1. We included this information in the results text section.

Results section first paragraph: 

“ The data from age and FVC (%pred.) are considered nonparametric and expressed as median and interquartile range [25% - 75%].”

Footnotes of the table: 

"Parametric data were presented as mean and standard deviation and nonparametric data as a median and interquartile range [25% - 75%]."

---

## [Editor Report · Decision Letter 2]

1 Jun 2023

Reliability of maximal respiratory nasal pressure tests in healthy young adults

PONE-D-22-15898R2

Dear Dr. Fregonezi,

We’re pleased to inform you that your manuscript has been judged scientifically suitable for publication and will be formally accepted for publication once it meets all outstanding technical requirements.

Kind regards,

Emiliano Cè

Academic Editor

PLOS ONE
---

## [Editor Report · Acceptance letter]

20 Jul 2023

PONE-D-22-15898R2 

Reliability of maximal respiratory nasal pressure tests in healthy young adults 

Dear Dr. Fregonezi:

I'm pleased to inform you that your manuscript has been deemed suitable for publication in PLOS ONE. Congratulations! Your manuscript is now with our production department. 

Kind regards, 

on behalf of

Prof. Emiliano Cè 

Academic Editor

PLOS ONE